# BERTology Meets Biology: Interpreting Attention in Protein Language Models

**Jesse Vig**[1]    **Ali Madani**[1]    **Lav R. Varshney**[1,2]    **Caiming Xiong**[1]
**Richard Socher**[1]    **Nazneen Fatema Rajani**[1]
[1]Salesforce Research, [2]University of Illinois at Urbana-Champaign
`{jvig,amadani,cxiong,rsocher,nazneen.rajani}@salesforce.com`
`varshney@illinois.edu`

## Abstract

Transformer architectures have proven to learn useful representations for protein classification and generation tasks. However, these representations present challenges in interpretability. In this work, we demonstrate a set of methods for analyzing protein Transformer models through the lens of attention. We show that attention: (1) captures the folding structure of proteins, connecting amino acids that are far apart in the underlying sequence, but spatially close in the three-dimensional structure, (2) targets binding sites, a key functional component of proteins, and (3) focuses on progressively more complex biophysical properties with increasing layer depth. We find this behavior to be consistent across three Transformer architectures (BERT, ALBERT, XLNet) and two distinct protein datasets. We also present a three-dimensional visualization of the interaction between attention and protein structure. Code for visualization and analysis is available at `https://github.com/salesforce/provis`.

## 1 Introduction

The study of proteins, the fundamental macromolecules governing biology and life itself, has led to remarkable advances in understanding human health and the development of disease therapies. The decreasing cost of sequencing technology has enabled vast databases of naturally occurring proteins (El-Gebali et al., 2019a), which are rich in information for developing powerful machine learning models of protein sequences. For example, sequence models leveraging principles of co-evolution, whether modeling pairwise or higher-order interactions, have enabled prediction of structure or function (Rollins et al., 2019).

Proteins, as a sequence of amino acids, can be viewed precisely as a language and therefore modeled using neural architectures developed for natural language. In particular, the Transformer (Vaswani et al., 2017), which has revolutionized unsupervised learning for text, shows promise for similar impact on protein sequence modeling. However, the strong performance of the Transformer comes at the cost of interpretability, and this lack of transparency can hide underlying problems such as model bias and spurious correlations (Niven & Kao, 2019; Tan & Celis, 2019; Kurita et al., 2019). In response, much NLP research now focuses on interpreting the Transformer, e.g., the subspecialty of "BERTology" (Rogers et al., 2020), which specifically studies the BERT model (Devlin et al., 2019).

In this work, we adapt and extend this line of interpretability research to protein sequences. We analyze Transformer protein models through the lens of attention, and present a set of interpretability methods that capture the unique functional and structural characteristics of proteins. We also compare the knowledge encoded in attention weights to that captured by hidden-state representations. Finally, we present a visualization of attention contextualized within three-dimensional protein structure.

Our analysis reveals that attention captures high-level structural properties of proteins, connecting amino acids that are spatially close in three-dimensional structure, but apart in the underlying sequence (Figure 1a). We also find that attention targets binding sites, a key functional component of proteins (Figure 1b). Further, we show how attention is consistent with a classic measure of similarity between amino acids—the substitution matrix. Finally, we demonstrate that attention captures progressively higher-level representations of structure and function with increasing layer depth.

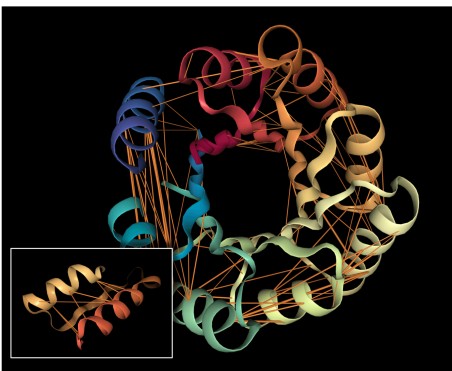

(a) Attention in head 12-4, which targets amino acid pairs that are close in physical space (see inset subsequence 117D-157I) but lie apart in the sequence. Example is a *de novo* designed TIM-barrel (5BVL) with characteristic symmetry.

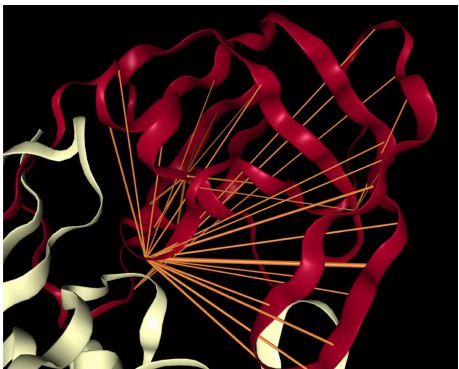

(b) Attention in head 7-1, which targets binding sites, a key functional component of proteins. Example is HIV-1 protease (7HVP). The primary location receiving attention is 27G, a binding site for protease inhibitor small-molecule drugs.

Figure 1: Examples of how specialized attention heads in a Transformer recover protein structure and function, based solely on language model pre-training. Orange lines depict attention between amino acids (line width proportional to attention weight; values below 0.1 hidden). Heads were selected based on correlation with ground-truth annotations of contact maps and binding sites. Visualizations based on the NGL Viewer (Rose et al., 2018; Rose & Hildebrand, 2015; Nguyen et al., 2017).

In contrast to NLP, which aims to automate a capability that humans already have—understanding natural language—protein modeling also seeks to shed light on biological processes that are not fully understood. Thus we also discuss how interpretability can aid scientific discovery.

## 2 BACKGROUND: PROTEINS

In this section we provide background on the biological concepts discussed in later sections.

**Amino acids.** Just as language is composed of words from a shared lexicon, every protein sequence is formed from a vocabulary of amino acids, of which 20 are commonly observed. Amino acids may be denoted by their full name (e.g., *Proline*), a 3-letter abbreviation (*Pro*), or a single-letter code (*P*).

**Substitution matrix.** While word synonyms are encoded in a thesaurus, proteins that are similar in structure or function are captured in a *substitution matrix*, which scores pairs of amino acids on how readily they may be substituted for one another while maintaining protein viability. One common substitution matrix is BLOSUM (Henikoff & Henikoff, 1992), which is derived from co-occurrence statistics of amino acids in aligned protein sequences.

**Protein structure.** Though a protein may be abstracted as a sequence of amino acids, it represents a physical entity with a well-defined three-dimensional structure (Figure 1). *Secondary structure* describes the local segments of proteins; two commonly observed types are the *alpha helix* and *beta sheet*. *Tertiary structure* encompasses the large-scale formations that determine the overall shape and function of the protein. One way to characterize tertiary structure is by a *contact map*, which describes the pairs of amino acids that are in contact (within 8 angstroms of one another) in the folded protein structure but lie apart (by at least 6 positions) in the underlying sequence (Rao et al., 2019).

**Binding sites.** Proteins may also be characterized by their functional properties. *Binding sites* are protein regions that bind with other molecules (proteins, natural ligands, and small-molecule drugs) to carry out a specific function. For example, the HIV-1 protease is an enzyme responsible for a critical process in replication of HIV (Brik & Wong, 2003). It has a binding site, shown in Figure 1b, that is a target for drug development to ensure inhibition.

**Post-translational modifications.** After a protein is translated from RNA, it may undergo additional modifications, e.g. phosphorylation, which play a key role in protein structure and function.

## 3 METHODOLOGY

**Model.** We demonstrate our interpretability methods on five Transformer models that were pretrained through language modeling of amino acid sequences. We primarily focus on the BERT-Base model from TAPE (Rao et al., 2019), which was pretrained on Pfam, a dataset of 31M protein sequences (El-Gebali et al., 2019b). We refer to this model as *TapeBert*. We also analyze 4 pre-trained Transformer models from ProtTrans (Elnaggar et al., 2020): *ProtBert* and *ProtBert-BFD*, which are 30-layer, 16-head BERT models; *ProtAlbert*, a 12-layer, 64-head ALBERT (Lan et al., 2020) model; and *ProtXLNet*, a 30-layer, 16-head XLNet (Yang et al., 2019) model. *ProtBert-BFD* was pretrained on BFD (Steinegger & Söding, 2018), a dataset of 2.1B protein sequences, while the other ProtTrans models were pretrained on UniRef100 (Suzek et al., 2014), which includes 216M protein sequences. A summary of these 5 models is presented in Appendix A.1.

Here we present an overview of BERT, with additional details on all models in Appendix A.2. BERT inputs a sequence of amino acids $\boldsymbol{x} = (x_1, \ldots, x_n)$ and applies a series of encoders. Each encoder layer $\ell$ outputs a sequence of continuous embeddings $(\mathbf{h}_1^{(\ell)}, \ldots, \mathbf{h}_n^{(\ell)})$ using a multi-headed attention mechanism. Each attention head in a layer produces a set of attention weights $\alpha$ for an input, where $\alpha_{i,j} > 0$ is the attention from token $i$ to token $j$, such that $\sum_j \alpha_{i,j} = 1$. Intuitively, attention weights define the influence of every token on the next layer's representation for the current token. We denote a particular head by *<layer>-<head_index>*, e.g. head *3-7* for the 3rd layer's 7th head.

**Attention analysis.** We analyze how attention aligns with various protein properties. For properties of token pairs, e.g. contact maps, we define an indicator function $f(i, j)$ that returns 1 if the property is present in token pair $(i, j)$ (e.g., if amino acids $i$ and $j$ are in contact), and 0 otherwise. We then compute the proportion of high-attention token pairs ($\alpha_{i,j} > \theta$) where the property is present, aggregated over a dataset $\boldsymbol{X}$:

$$p_\alpha(f) = \sum_{\mathbf{x} \in \mathbf{X}} \sum_{i=1}^{|\mathbf{x}|} \sum_{j=1}^{|\mathbf{x}|} f(i, j) \cdot \mathbb{1}_{\alpha_{i,j} > \theta} \Bigg/ \sum_{\mathbf{x} \in \mathbf{X}} \sum_{i=1}^{|\mathbf{x}|} \sum_{j=1}^{|\mathbf{x}|} \mathbb{1}_{\alpha_{i,j} > \theta} \tag{1}$$

where $\theta$ is a threshold to select for high-confidence attention weights. We also present an alternative, continuous version of this metric in Appendix B.1.

For properties of *individual tokens*, e.g. binding sites, we define $f(i, j)$ to return 1 if the property is present in token $j$ (e.g. if $j$ is a binding site). In this case, $p_\alpha(f)$ equals the proportion of attention that is directed *to* the property (e.g. the proportion of attention focused on binding sites).

When applying these metrics, we include two types of checks to ensure that the results are not due to chance. First, we test that the proportion of attention that aligns with particular properties is significantly higher than the background frequency of these properties, taking into account the Bonferroni correction for multiple hypotheses corresponding to multiple attention heads. Second, we compare the results to a null model, which is an instance of the model with randomly shuffled attention weights. We describe these methods in detail in Appendix B.2.

**Probing tasks.** We also perform *probing tasks* on the model, which test the knowledge contained in model representations by using them as inputs to a classifier that predicts a property of interest (Veldhoen et al., 2016; Conneau et al., 2018; Adi et al., 2016). The performance of the probing classifier serves as a measure of the knowledge of the property that is encoded in the representation. We run both *embedding probes*, which assess the knowledge encoded in the output embeddings of each layer, and *attention probes* (Reif et al., 2019; Clark et al., 2019), which measure the knowledge contained in the attention weights for pairwise features. Details are provided in Appendix B.3.

**Datasets.** For our analyses of amino acids and contact maps, we use a curated dataset from TAPE based on ProteinNet (AlQuraishi, 2019; Fox et al., 2013; Berman et al., 2000; Moult et al., 2018), which contains amino acid sequences annotated with spatial coordinates (used for the contact map analysis). For the analysis of secondary structure and binding sites we use the Secondary Structure dataset (Rao et al., 2019; Berman et al., 2000; Moult et al., 2018; Klausen et al., 2019) from TAPE. We employed a taxonomy of secondary structure with three categories: *Helix*, *Strand*, and *Turn/Bend*, with the last two belonging to the higher-level *beta sheet* category (Sec. 2). We used this taxonomy to study how the model understood structurally distinct regions of beta sheets. We obtained token-level binding site and protein modification labels from the Protein Data Bank (Berman et al., 2000). For analyzing attention, we used a random subset of 5000 sequences from the training split of the

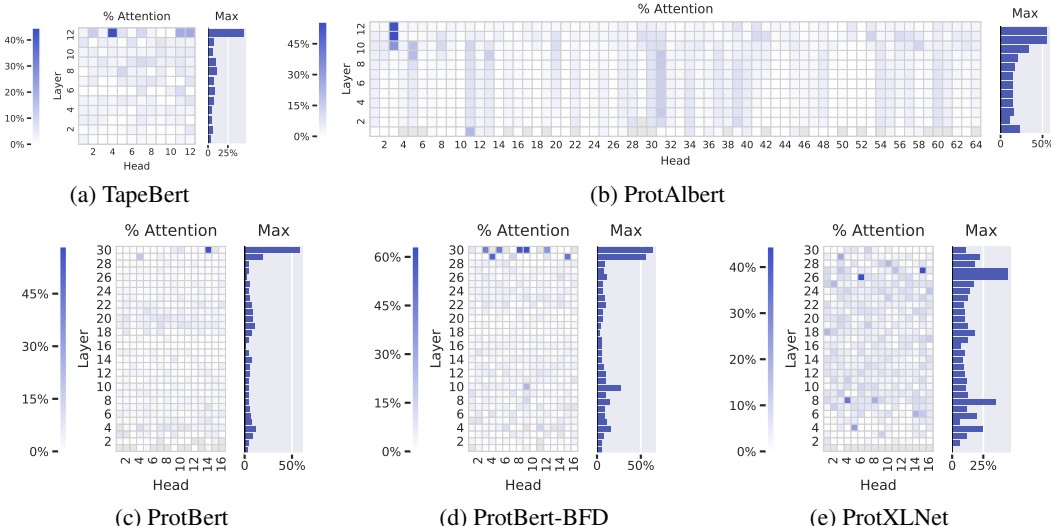

Figure 2: Agreement between attention and contact maps across five pretrained Transformer models from TAPE (a) and ProtTrans (b–e). The heatmaps show the proportion of high-confidence attention weights ($\alpha_{i,j} > \theta$) from each head that connects pairs of amino acids that are in contact with one another. In TapeBert (a), for example, we can see that 45% of attention in head 12-4 (the 12th layer's 4th head) maps to contacts. The bar plots show the maximum value from each layer. Note that the vertical striping in ProtAlbert (b) is likely due to cross-layer parameter sharing (see Appendix A.3).

respective datasets (note that none of the aforementioned annotations were used in model training). For the diagnostic classifier, we used the respective training splits for training and the validation splits for evaluation. See Appendix B.4 for additional details.

**Experimental details** We exclude attention to the `[SEP]` delimiter token, as it has been shown to be a "no-op" attention token (Clark et al., 2019), as well as attention to the `[CLS]` token, which is not explicitly used in language modeling. We only include results for attention heads where at least 100 high-confidence attention arcs are available for analysis. We set the attention threshold $\theta$ to 0.3 to select for high-confidence attention while retaining sufficient data for analysis. We truncate all protein sequences to a length of 512 to reduce memory requirements.[1]

We note that all of the above analyses are purely associative and do not attempt to establish a causal link between attention and model behavior (Vig et al., 2020; Grimsley et al., 2020), nor to *explain* model predictions (Jain & Wallace, 2019; Wiegreffe & Pinter, 2019).

## 4    WHAT DOES ATTENTION UNDERSTAND ABOUT PROTEINS?

### 4.1    PROTEIN STRUCTURE

Here we explore the relationship between attention and tertiary structure, as characterized by contact maps (see Section 2). Secondary structure results are included in Appendix C.1.

**Attention aligns strongly with contact maps in the deepest layers.** Figure 2 shows how attention aligns with contact maps across the heads of the five models evaluated[2], based on the metric defined in Equation 1. The most aligned heads are found in the deepest layers and focus up to 44.7% (TapeBert), 55.7% (ProtAlbert), 58.5% (ProtBert), 63.2% (ProtBert-BFD), and 44.5% (ProtXLNet) of attention on contacts, whereas the background frequency of contacts among all amino acid pairs in the dataset is 1.3%. Figure 1a shows an example of the induced attention from the top head in TapeBert. We note that the model with the single most aligned head—ProtBert-BFD—is the largest model (same size as ProteinBert) at 420M parameters (Appendix A.1) and it was also the only model pre-trained on the

---

[1]94% of sequences had length less than 512. Experiments performed on single 16GB Tesla V-100 GPU.
[2]Heads with fewer than 100 high-confidence attention weights across the dataset are grayed out.

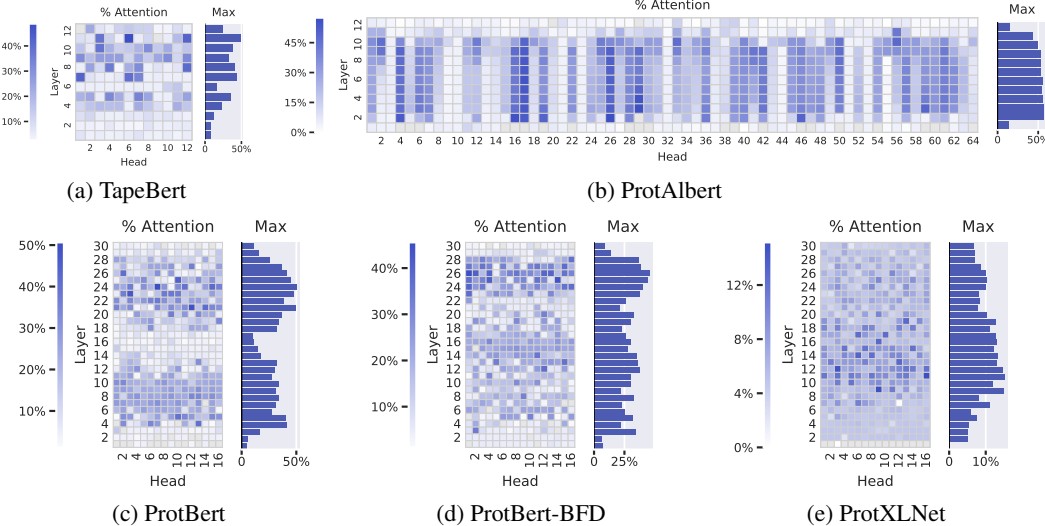

Figure 3: Proportion of attention focused on binding sites across five pretrained models. The heatmaps show the proportion of high-confidence attention ($\alpha_{i,j} > \theta$) from each head that is directed to binding sites. In TapeBert (a), for example, we can see that 49% of attention in head 11-6 (the 11th layer's 6th head) is directed to binding sites. The bar plots show the maximum value from each layer.

largest dataset, BFD. It's possible that both factors helped the model learn more structurally-aligned attention patterns. Statistical significance tests and null models are reported in Appendix C.2.

Considering the models were trained on language modeling tasks without any spatial information, the presence of these structurally-aware attention heads is intriguing. One possible reason for this emergent behavior is that contacts are more likely to biochemically interact with one another, creating statistical dependencies between the amino acids in contact. By focusing attention on the contacts of a masked position, the language models may acquire valuable context for token prediction.

While there seems to be a strong correlation between the attention head output and classically-defined contacts, there are also differences. The models may have learned differing contextualized or nuanced formulations that describe amino acid interactions. These learned interactions could then be used for further discovery and investigation or repurposed for prediction tasks similar to how principles of coevolution enabled a powerful representation for structure prediction.

### 4.2 BINDING SITES AND POST-TRANSLATIONAL MODIFICATIONS

We also analyze how attention interacts with binding sites and post-translational modifications (PTMs), which both play a key role in protein function.

**Attention targets binding sites throughout most layers of the models.** Figure 3 shows the proportion of attention focused on binding sites (Eq. 1) across the heads of the 5 models studied. Attention to binding sites is most pronounced in the ProtAlbert model (Figure 3b), which has 22 heads that focus over 50% of attention on bindings sites, whereas the background frequency of binding sites in the dataset is 4.8%. The three BERT models (Figures 3a, 3c, and 3d) also attend strongly to binding sites, with attention heads focusing up to 48.2%, 50.7%, and 45.6% of attention on binding sites, respectively. Figure 1b visualizes the attention in one strongly-aligned head from the TapeBert model. Statistical significance tests and a comparison to a null model are provided in Appendix C.3.

ProtXLNet (Figure 3e) also targets binding sites, but not as strongly as the other models: the most aligned head focuses 15.1% of attention on binding sites, and the average head directs just 6.2% of attention to binding sites, compared to 13.2%, 19.8%, 16.0%, and 15.1% for the first four models in Figure 3. It's unclear whether this disparity is due to differences in architectures or pre-training objectives; for example, ProtXLNet uses a bidirectional auto-regressive pretraining method (see Appendix A.2), whereas the other 4 models all use masked language modeling objectives.

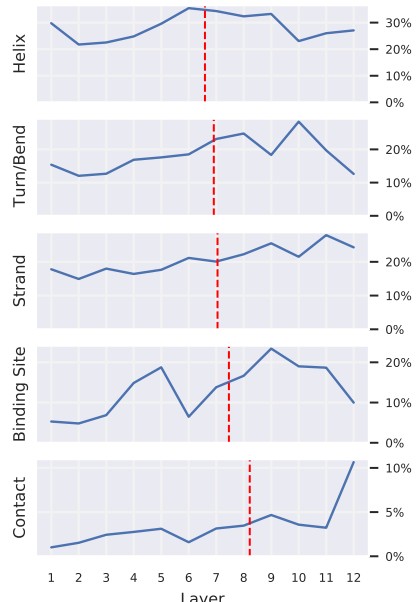
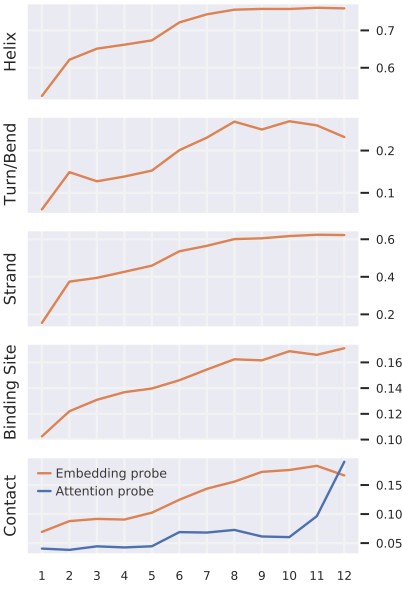

Figure 4: Each plot shows the percentage of attention focused on the given property, averaged over all heads within each layer. The plots, sorted by center of gravity (red dashed line), show that heads in deeper layers focus relatively more attention on binding sites and contacts, whereas attention toward specific secondary structures is more even across layers.

Figure 5: Performance of probing classifiers by layer, sorted by task order in Figure 4. The embedding probes (orange) quantify the knowledge of the given property that is encoded in each layer's output embeddings. The attention probe (blue), show the amount of information encoded in attention weights for the (pairwise) contact feature. Additional details are provided in Appendix B.3.

Why does attention target binding sites? In contrast to contact maps, which reveal relationships *within* proteins, binding sites describe how a protein interacts with other molecules. These external interactions ultimately define the high-level function of the protein, and thus binding sites remain conserved even when the sequence as a whole evolves (Kinjo & Nakamura, 2009). Further, structural motifs in binding sites are mainly restricted to specific families or superfamilies of proteins (Kinjo & Nakamura, 2009), and binding sites can reveal evolutionary relationships among proteins (Lee et al., 2017). Thus binding sites may provide the model with a high-level characterization of the protein that is robust to individual sequence variation. By attending to these regions, the model can leverage this higher-level context when predicting masked tokens throughout the sequence.

**Attention targets PTMs in a small number of heads.** A small number of heads in each model concentrate their attention very strongly on amino acids associated with post-translational modifications (PTMs). For example, Head 11-6 in TapeBert focused 64% of attention on PTM positions, though these occur at only 0.8% of sequence positions in the dataset.[3] Similar to our discussion on binding sites, PTMs are critical to protein function (Rubin & Rosen, 1975) and thereby are likely to exhibit behavior that is conserved across the sequence space. See Appendix C.4 for full results.

### 4.3 CROSS-LAYER ANALYSIS

We analyze how attention captures properties of varying complexity across different layers of TapeBert, and compare this to a probing analysis of embeddings and attention weights (see Section 3).

**Attention targets higher-level properties in deeper layers.** As shown in Figure 4, deeper layers focus relatively more attention on binding sites and contacts (high-level concept), whereas secondary structure (low- to mid-level concept) is targeted more evenly across layers. The probing analysis of attention (Figure 5, blue) similarly shows that knowledge of contact maps (a pairwise feature)

---

[3]This head also targets binding sites (Fig. 3a) but at a percentage of 49%.

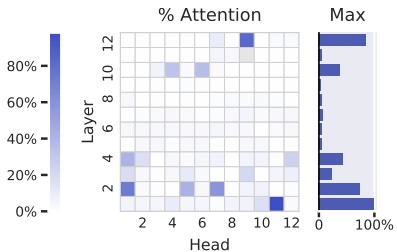 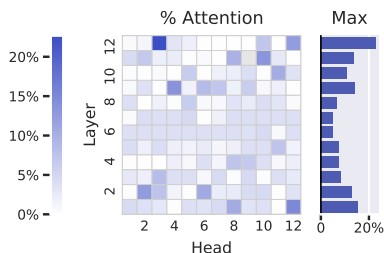

Figure 6: Percentage of each head's attention focused on amino acids *Pro* (left) and *Phe* (right).

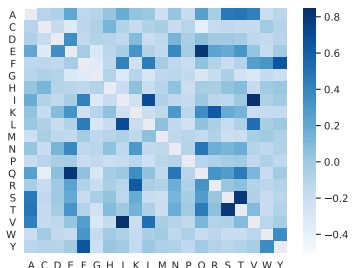 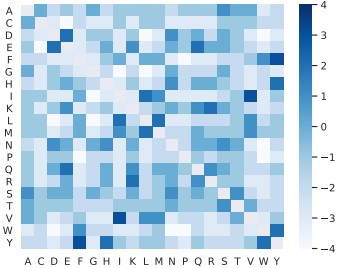

Figure 7: Pairwise attention similarity (left) vs. substitution matrix (right) (codes in App. C.5)

is encoded in attention weights primarily in the last 1-2 layers. These results are consistent with prior work in NLP that suggests deeper layers in text-based Transformers attend to more complex properties (Vig & Belinkov, 2019) and encode higher-level representations (Raganato & Tiedemann, 2018; Peters et al., 2018; Tenney et al., 2019; Jawahar et al., 2019).

The *embedding* probes (Figure 5, orange) also show that the model first builds representations of local secondary structure in lower layers before fully encoding binding sites and contact maps in deeper layers. However, this analysis also reveals stark differences in how knowledge of contact maps is accrued in embeddings, which accumulate this knowledge gradually over many layers, compared to attention weights, which acquire this knowledge only in the final layers in this case. This example points out limitations of common layerwise probing approaches that only consider embeddings, which, intuitively, represent what the model *knows* but not necessarily how it operationalizes that knowledge.

## 4.4 AMINO ACIDS AND THE SUBSTITUTION MATRIX

In addition to high-level structural and functional properties, we also performed a fine-grained analysis of the interaction between attention and particular amino acids.

**Attention heads specialize in particular amino acids.** We computed the proportion of TapeBert's attention to each of the 20 standard amino acids, as shown in Figure 6 for two example amino acids. For 16 of the amino acids, there exists an attention head that focuses over 25% of attention on that amino acid, significantly greater than the background frequencies of the corresponding amino acids, which range from 1.3% to 9.4%. Similar behavior was observed for ProtBert, ProtBert-BFD, ProtAlbert, and ProtXLNet models, with 17, 15, 16, and 18 amino acids, respectively, receiving greater than 25% of the attention from at least one attention head. Detailed results for TapeBert including statistical significance tests and comparison to a null model are presented in Appendix C.5.

**Attention is consistent with substitution relationships.** A natural follow-up question from the above analysis is whether each head has "memorized" specific amino acids to target, or whether it has actually learned meaningful properties that correlate with particular amino acids. To test the latter hypothesis, we analyze whether amino acids with similar structural and functional properties are attended to similarly across heads. Specifically, we compute the Pearson correlation between the distribution of attention across heads between all pairs of distinct amino acids, as shown in Figure 7 (left) for TapeBert. For example, the entry for *Pro* (P) and *Phe* (F) is the correlation between the two heatmaps in Figure 6. We compare these scores to the BLOSUM62 substitution scores (Sec. 2) in Figure 7 (right), and find a Pearson correlation of 0.73, suggesting that attention is moderately

consistent with substitution relationships. Similar correlations are observed for the ProtTrans models: 0.68 (ProtBert), 0.75 (ProtBert-BFD), 0.60 (ProtAlbert), and 0.71 (ProtXLNet). As a baseline, the randomized versions of these models (Appendix B.2) yielded correlations of -0.02 (TapeBert), 0.02 (ProtBert), -0.03 (ProtBert-BFD), -0.05 (ProtAlbert), and 0.21 (ProtXLNet).

## 5 RELATED WORK

### 5.1 PROTEIN LANGUAGE MODELS

Deep neural networks for protein language modeling have received broad interest. Early work applied the Skip-gram model (Mikolov et al., 2013) to construct continuous embeddings from protein sequences (Asgari & Mofrad, 2015). Sequence-only language models have since been trained through autoregressive or autoencoding self-supervision objectives for discriminative and generative tasks, for example, using LSTMs or Transformer-based architectures (Alley et al., 2019; Bepler & Berger, 2019; Rao et al., 2019; Rives et al., 2019). TAPE created a benchmark of five tasks to assess protein sequence models, and ProtTrans also released several large-scale pretrained protein Transformer models (Elnaggar et al., 2020). Riesselman et al. (2019); Madani et al. (2020) trained autoregressive generative models to predict the functional effect of mutations and generate natural-like proteins.

From an interpretability perspective, Rives et al. (2019) showed that the output embeddings from a pretrained Transformer can recapitulate structural and functional properties of proteins through learned linear transformations. Various works have analyzed output embeddings of protein models through dimensionality reduction techniques such as PCA or t-SNE (Elnaggar et al., 2020; Biswas et al., 2020). In our work, we take an interpretability-first perspective to focus on the internal model representations, specifically attention and intermediate hidden states, across multiple protein language models. We also explore novel biological properties including binding sites and post-translational modifications.

### 5.2 INTERPRETING MODELS IN NLP

The rise of deep neural networks in ML has also led to much work on interpreting these so-called black-box models. This section reviews the NLP interpretability literature on the Transformer model, which is directly comparable to our work on interpreting Transformer models of protein sequences.

**Interpreting Transformers.** The Transformer is a neural architecture that uses attention to accelerate learning (Vaswani et al., 2017). In NLP, transformers are the backbone of state-of-the-art pre-trained language models such as BERT (Devlin et al., 2019). *BERTology* focuses on interpreting what the BERT model learns about language using a suite of probes and interventions (Rogers et al., 2020). So-called *diagnostic classifiers* are used to interpret the outputs from BERT's layers (Veldhoen et al., 2016). At a high level, mechanisms for interpreting BERT can be placed into three main categories: interpreting the learned embeddings (Ethayarajh, 2019; Wiedemann et al., 2019; Mickus et al., 2020; Adi et al., 2016; Conneau et al., 2018), BERT's learned knowledge of syntax (Lin et al., 2019; Liu et al., 2019; Tenney et al., 2019; Htut et al., 2019; Hewitt & Manning, 2019; Goldberg, 2019), and BERT's learned knowledge of semantics (Tenney et al., 2019; Ettinger, 2020).

**Interpreting attention specifically.** Interpreting attention on textual sequences is a well-established area of research (Wiegreffe & Pinter, 2019; Zhong et al., 2019; Brunner et al., 2020; Hewitt & Manning, 2019). Past work has been shown that attention correlates with syntactic and semantic relationships in natural language in some cases (Clark et al., 2019; Vig & Belinkov, 2019; Htut et al., 2019). Depending on the task and model architecture, attention may have less or more explanatory power for model predictions (Jain & Wallace, 2019; Serrano & Smith, 2019; Pruthi et al., 2020; Moradi et al., 2019; Vashishth et al., 2019). Visualization techniques have been used to convey the structure and properties of attention in Transformers (Vaswani et al., 2017; Kovaleva et al., 2019; Hoover et al., 2020; Vig, 2019). Recent work has begun to analyze attention in Transformer models outside of the domain of natural language (Schwaller et al., 2020; Payne et al., 2020).

Our work extends these methods to protein sequence models by considering particular biophysical properties and relationships. We also present a joint cross-layer probing analysis of attention weights and layer embeddings. While past work in NLP has analyzed attention and embeddings across layers, we believe we are the first to do so in any domain using a single, unified metric, which enables us to

directly compare the relative information content of the two representations. Finally, we present a novel tool for visualizing attention embedded in three-dimensional structure.

## 6 CONCLUSIONS AND FUTURE WORK

This paper builds on the synergy between NLP and computational biology by adapting and extending NLP interpretability methods to protein sequence modeling. We show how a Transformer language model recovers structural and functional properties of proteins and integrates this knowledge directly into its attention mechanism. While this paper focuses on reconciling attention with known properties of proteins, one might also leverage attention to uncover novel relationships or more nuanced forms of existing measures such as contact maps, as discussed in Section 4.1. In this way, language models have the potential to serve as tools for scientific discovery. But in order for learned representations to be accessible to domain experts, they must be presented in an appropriate context to facilitate discovery. Visualizing attention in the context of protein structure (Figure 1) is one attempt to do so. We believe there is the potential to develop such contextual visualizations of learned representations in a range of scientific domains.

## ACKNOWLEDGMENTS

We would like to thank Xi Victoria Lin, Stephan Zheng, Melvin Gruesbeck, and the anonymous reviewers for their valuable feedback.

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

# A  MODEL OVERVIEW

## A.1  PRE-TRAINED MODELS

Table 1 provides an overview of the five pre-trained Transformer models studied in this work. The models originate from the TAPE and ProtTrans repositories, spanning three model architectures: BERT, ALBERT, and XLNet.

Table 1: Summary of pre-trained models analyzed, including the source of the model, the type of Transformer used, the number of layers and heads, the total number of model parameters, the source of the pre-training dataset, and the number of protein sequences in the pre-training dataset.

| Source | Name | Type | Layers | Heads | Params | Train Dataset | # Seq |
|---|---|---|---|---|---|---|---|
| TAPE | TapeBert | BERT | 12 | 12 | 94M | Pfam | 31M |
| ProtTrans | ProtBert | BERT | 30 | 16 | 420M | Uniref100 | 216M |
| ProtTrans | ProtBert-BFD | BERT | 30 | 16 | 420M | BFD | 2.1B |
| ProtTrans | ProtAlbert | ALBERT | 12 | 64 | 224M | Uniref100 | 216M |
| ProtTrans | ProtXLNet | XLNet | 30 | 16 | 409M | Uniref100 | 216M |

## A.2  BERT TRANSFORMER ARCHITECTURE

**Stacked Encoder:**  BERT uses a stacked-encoder architecture, which inputs a sequence of tokens $x = (x_1, ..., x_n)$ and applies position and token embeddings followed by a series of encoder layers. Each layer applies multi-head self-attention (see below) in combination with a feedforward network, layer normalization, and residual connections. The output of each layer $\ell$ is a sequence of contextualized embeddings $(\mathbf{h}_1^{(\ell)}, \ldots, \mathbf{h}_n^{(\ell)})$.

**Self-Attention:**  Given an input $x = (x_1, \ldots, x_n)$, the self-attention mechanism assigns to each token pair $i, j$ an attention weight $\alpha_{i,j} > 0$ where $\sum_j \alpha_{i,j} = 1$. Attention in BERT is bidirectional. In the multi-layer, multi-head setting, $\alpha$ is specific to a layer and head. The BERT-Base model has 12 layers and 12 heads. Each attention head learns a distinct set of weights, resulting in 12 x 12 = 144 distinct attention mechanisms in this case.

The attention weights $\alpha_{i,j}$ are computed from the scaled dot-product of the *query vector* of $i$ and the *key vector* of $j$, followed by a softmax operation. The attention weights are then used to produce a weighted sum of value vectors:

$$\text{Attention}(Q, K, V) = \text{softmax}\left(\frac{QK^T}{\sqrt{d_k}}\right) V \tag{2}$$

using query matrix $Q$, key matrix $K$, and value matrix $V$, where $d_k$ is the dimension of $K$. In a multi-head setting, the queries, keys, and values are linearly projected $h$ times, and the attention operation is performed in parallel for each representation, with the results concatenated.

## A.3  OTHER TRANSFORMER VARIANTS

**ALBERT:**  The architecture of ALBERT differs from BERT in two ways: (1) It shares parameters across layers, unlike BERT which learns distinct parameters for every layer and (2) It uses factorized embeddings, which allows the input token embeddings to be of a different (smaller) size than the hidden states. The original version of ALBERT designed for text also employed a sentence-order prediction pretraining task, but this was not used on the models studied in this paper.

**XLNet:**  Instead of the masked-language modeling pretraining objective use for BERT, XLNet uses a bidirectional auto-regressive pretraining method that considers all possible orderings of the input factorization. The architecture also adds a segment recurrence mechanism to process long sequences, as well as a relative rather than absolute encoding scheme.

# B    ADDITIONAL EXPERIMENTAL DETAILS

## B.1    ALTERNATIVE ATTENTION AGREEMENT METRIC

Here we present an alternative formulation to Eq. 1 based on an attention-weighted average. We define an indicator function $f(i, j)$ for property $f$ that returns 1 if the property is present in token pair $(i, j)$ (i.e., if amino acids $i$ and $j$ are in contact), and zero otherwise. We then compute the proportion of attention that matches with $f$ over a dataset $X$ as follows:

$$p_\alpha(f) = \sum_{x \in X} \sum_{i=1}^{|x|} \sum_{j=1}^{|x|} f(i,j)\alpha_{i,j}(x) \bigg/ \sum_{x \in X} \sum_{i=1}^{|x|} \sum_{j=1}^{|x|} \alpha_{i,j}(x) \qquad (3)$$

where $\alpha_{i,j}(x)$ denotes the attention from $i$ to $j$ for input sequence $x$.

## B.2    STATISTICAL SIGNIFICANCE TESTING AND NULL MODELS

We perform statistical significance tests to determine whether any results based on the metric defined in Equation 1 are due to chance. Given a property $f$, as defined in Section 3, we perform a two-proportion z-test comparing (1) the proportion of high-confidence attention arcs ($\alpha_{i,j} > \theta$) for which $f(i, j) = 1$, and (2) the proportion of all possible pairs $i, j$ for which $f(i, j) = 1$. Note that the first proportion is exactly the metric $p_\alpha(f)$ defined in Equation 1 (e.g. the proportion of attention aligned with contact maps). The second proportion is simply the background frequency of the property (e.g. the background frequency of contacts). Since we extract the maximum scores over all of the heads in the model, we treat this as a case of multiple hypothesis testing and apply the Bonferroni correction, with the number of hypotheses $m$ equal to the number of attention heads.

As an additional check that the results did not occur by chance, we also report results on baseline (null) models. We initially considered using two forms of null models: (1) a model with randomly initialized weights. and (2) a model trained on randomly shuffled sequences. However, in both cases, none of the sequences in the dataset yielded attention weights greater than the attention threshold $\theta$. This suggests that the mere existence of the high-confidence attention weights used in the analysis could not have occurred by chance, but it does not shed light on the particular analyses performed. Therefore, we implemented an alternative randomization scheme in which we randomly shuffle attention weights from the original models as a post-processing step. Specifically, we permute the sequence of attention weights *from* each token for every attention head. To illustrate, let's say that the original model produced attention weights of (0.3, 0.2, 0.1, 0.4, 0.0) from position $i$ in protein sequence $x$ from head $h$, where $|x| = 5$. In the null model, the attention weights from position $i$ in sequence $x$ in head $h$ would be a random permutation of those weights, e.g., (0.2, 0.0, 0.4, 0.3, 0.1). Note that these are still valid attention weights as they would sum to 1 (since the original weights would sum to 1 by definition). We report results using this form of baseline model.

## B.3    PROBING METHODOLOGY

**Embedding probe.** We probe the embedding vectors output from each layer using a linear probing classifier. For token-level probing tasks (binding sites, secondary structure) we feed each token's output vector directly to the classifier. For token-pair probing tasks (contact map) we construct a pairwise feature vector by concatenating the elementwise differences and products of the two tokens' output vectors, following the TAPE[4] implementation.

We use task-specific evaluation metrics for the probing classifier: for secondary structure prediction, we measure F1 score; for contact prediction, we measure precision@$L/5$, where $L$ is the length of the protein sequence, following standard practice (Moult et al., 2018); for binding site prediction, we measure precision@$L/20$, since approximately one in twenty amino acids in each sequence is a binding site (4.8% in the dataset).

**Attention probe.** Just as the attention weight $\alpha_{i,j}$ is defined for a pair of amino acids $(i, j)$, so is the contact property $f(i, j)$, which returns true if amino acids $i$ and $j$ are in contact. Treating the attention weight as a feature of a token-pair $(i, j)$, we can train a probing classifier that predicts the

---

[4]https://github.com/songlab-cal/tape

contact property based on this feature, thereby quantifying the attention mechanism's knowledge of that property. In our multi-head setting, we treat the attention weights across all heads in a given layer as a feature vector, and use a probing classifier to assess the knowledge of a given property in the attention weights across the entire layer. As with the embedding probe, we measure performance of the probing classifier using precision@$L/5$, where $L$ is the length of the protein sequence, following standard practice for contact prediction.

## B.4 DATASETS

We used two protein sequence datasets from the TAPE repository for the analysis: the ProteinNet dataset (AlQuraishi, 2019; Fox et al., 2013; Berman et al., 2000; Moult et al., 2018) and the Secondary Structure dataset (Rao et al., 2019; Berman et al., 2000; Moult et al., 2018; Klausen et al., 2019). The former was used for analysis of amino acids and contact maps, and the latter was used for analysis of secondary structure. We additionally created a third dataset for binding site and post-translational modification (PTM) analysis from the Secondary Structure dataset, which was augmented with binding site and PTM annotations obtained from the Protein Data Bank's Web API.[5] We excluded any sequences for which annotations were not available. The resulting dataset sizes are shown in Table 2. For the analysis of attention, a random subset of 5000 sequences from the training split of each dataset was used, as the analysis was purely evaluative. For training and evaluating the diagnostic classifier, the full training and validation splits were used.

Table 2: Datasets used in analysis

| Dataset | Train size | Validation size |
|---|---|---|
| ProteinNet | 25299 | 224 |
| Secondary Structure | 8678 | 2170 |
| Binding Sites / PTM | 5734 | 1418 |

## C ADDITIONAL RESULTS OF ATTENTION ANALYSIS

### C.1 SECONDARY STRUCTURE

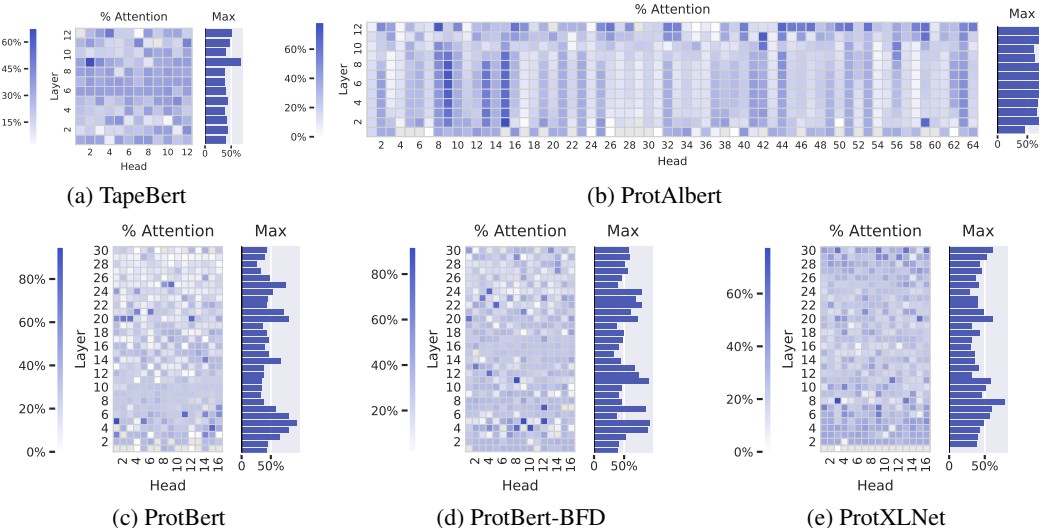

Figure 8: Percentage of each head's attention that is focused on *Helix* secondary structure.

---

[5] http://www.rcsb.org/pdb/software/rest.do

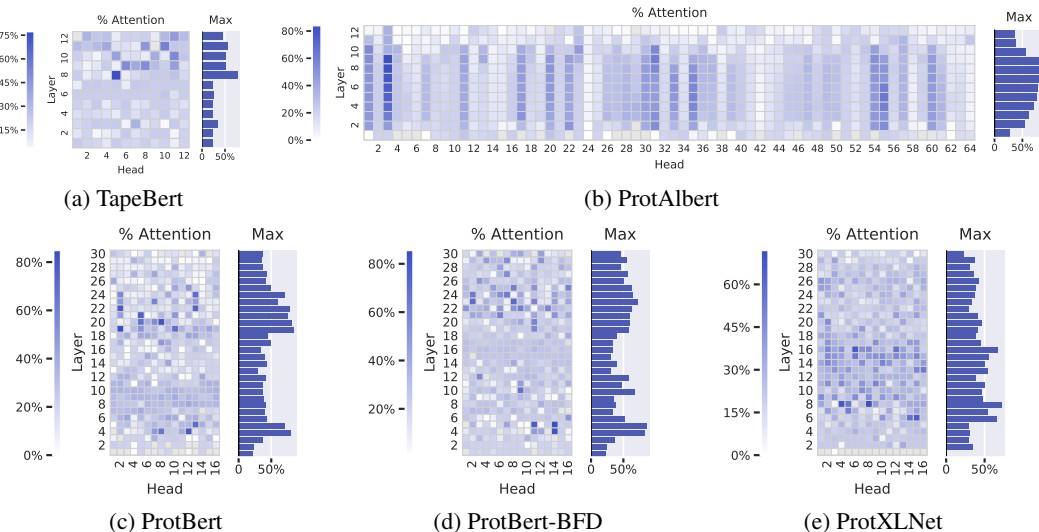

Figure 9: Percentage of each head's attention that is focused on *Strand* secondary structure.

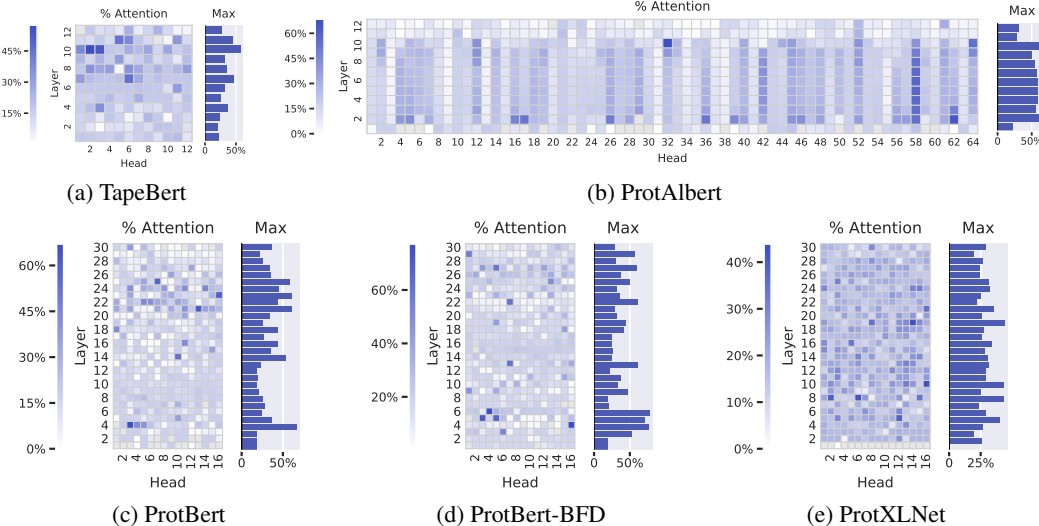

Figure 10: Percentage of each head's attention that is focused on *Turn/Bend* secondary structure.

## C.2 CONTACT MAPS: STATISTICAL SIGNIFICANCE TESTS AND NULL MODELS

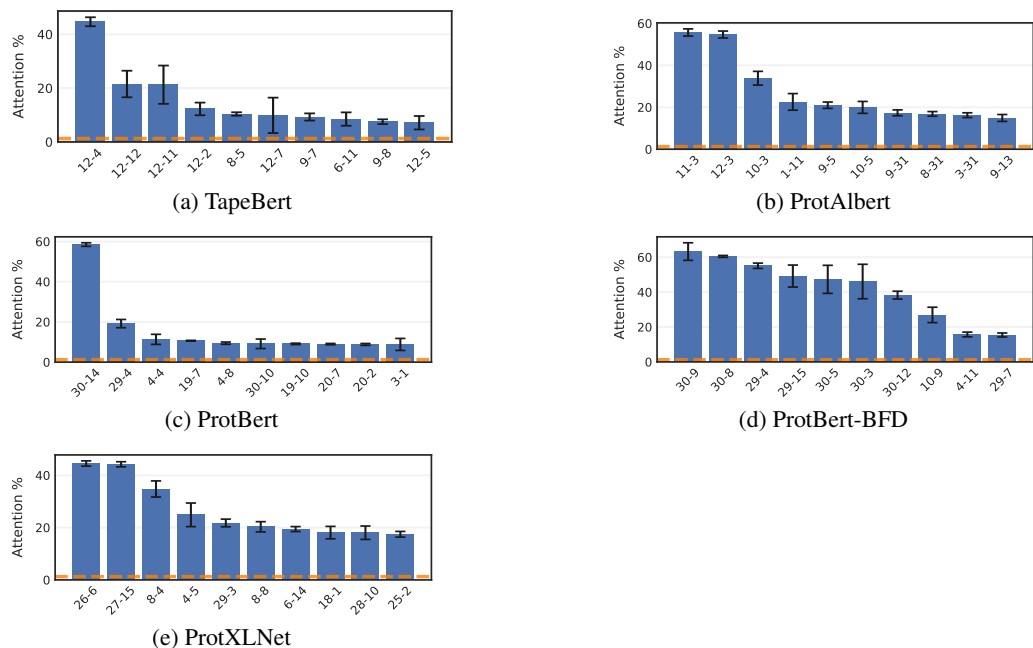

Figure 11: Top 10 heads (denoted by *<layer>-<head>*) for each model based on the proportion of attention aligned with contact maps [95% conf. intervals]. The differences between the attention proportions and the background frequency of contacts (orange dashed line) are statistically significant ($p < 0.00001$). Bonferroni correction applied for both confidence intervals and tests (see App. B.2).

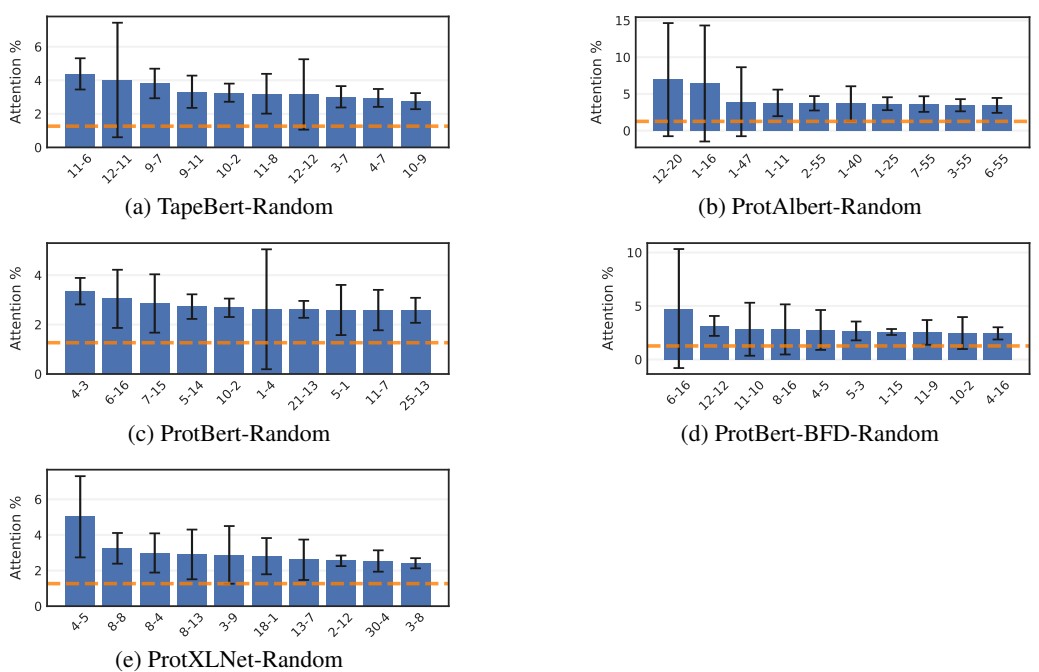

Figure 12: Top-10 contact-aligned heads for **null** models. See Appendix B.2 for details.

## C.3    BINDING SITES: STATISTICAL SIGNIFICANCE TESTS AND NULL MODEL

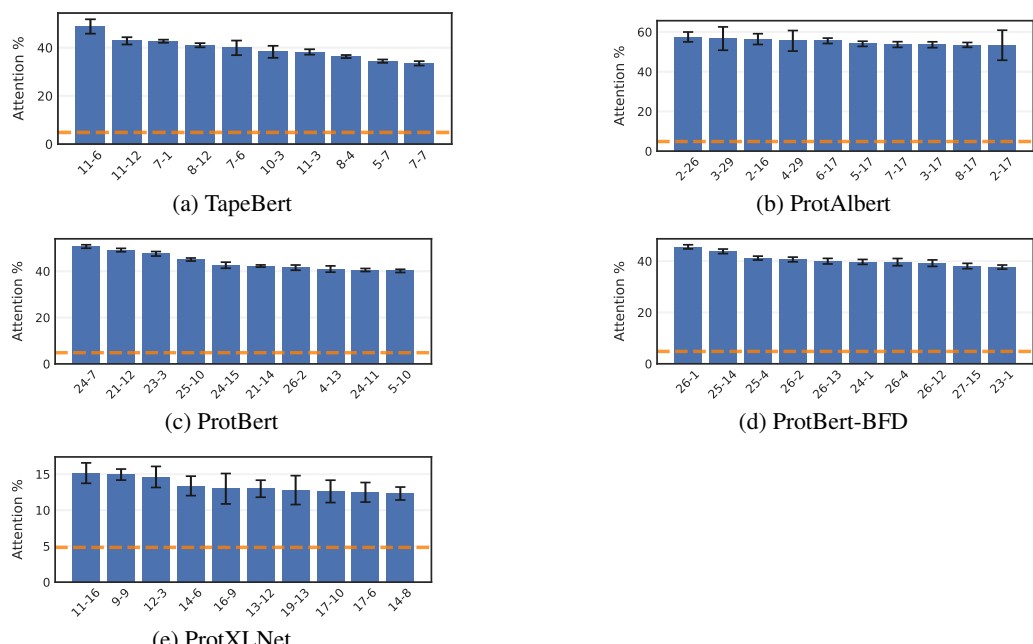

Figure 13: Top 10 heads (denoted by *<layer>-<head>*) for each model based on the proportion of attention focused on binding sites [95% conf. intervals]. Differences between attention proportions and the background frequency of binding sites (orange dashed line) are all statistically significant ($p < 0.00001$). Bonferroni correction applied for both confidence intervals and tests (see App. B.2).

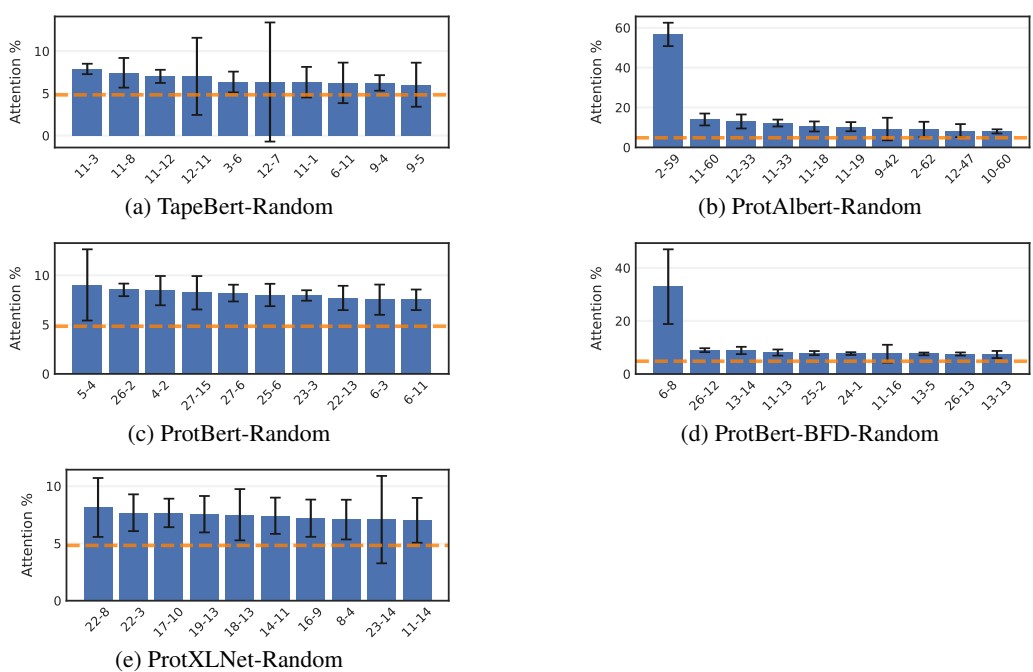

Figure 14: Top-10 heads most focused on binding sites for **null** models. See Appendix B.2 for details.

### C.4    POST-TRANSLATIONAL MODIFICATIONS (PTMs)

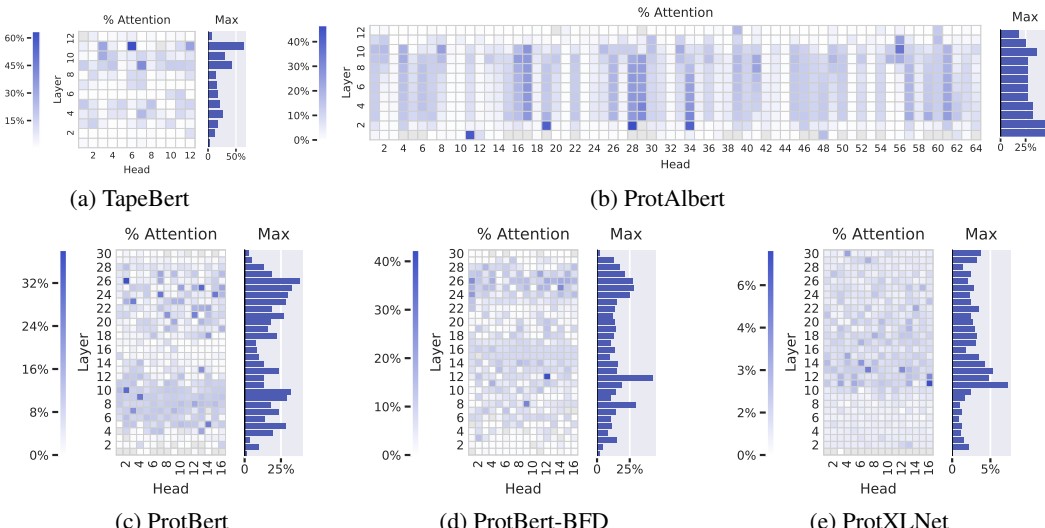

Figure 15: Percentage of each head's attention that is focused on post-translational modifications.

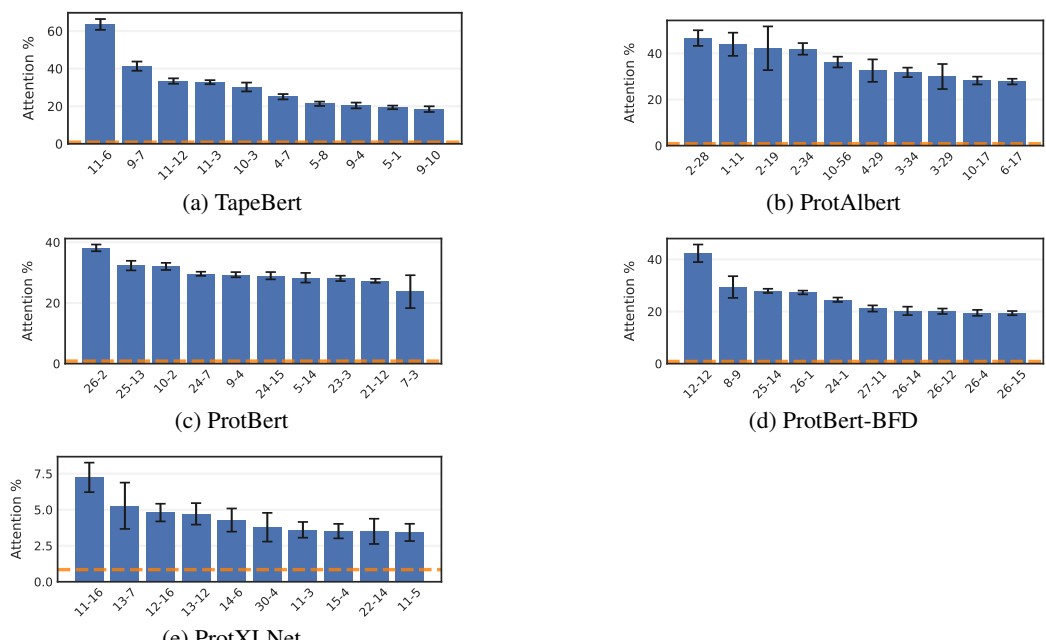

Figure 16: Top 10 heads (denoted by *<layer>-<head>*) for each model based on the proportion of attention focused on PTM positions [95% conf. intervals]. The differences between the attention proportions and the background frequency of PTMs (orange dashed line) are statistically significant ($p < 0.00001$). Bonferroni correction applied for both confidence intervals and tests (see App. B.2).

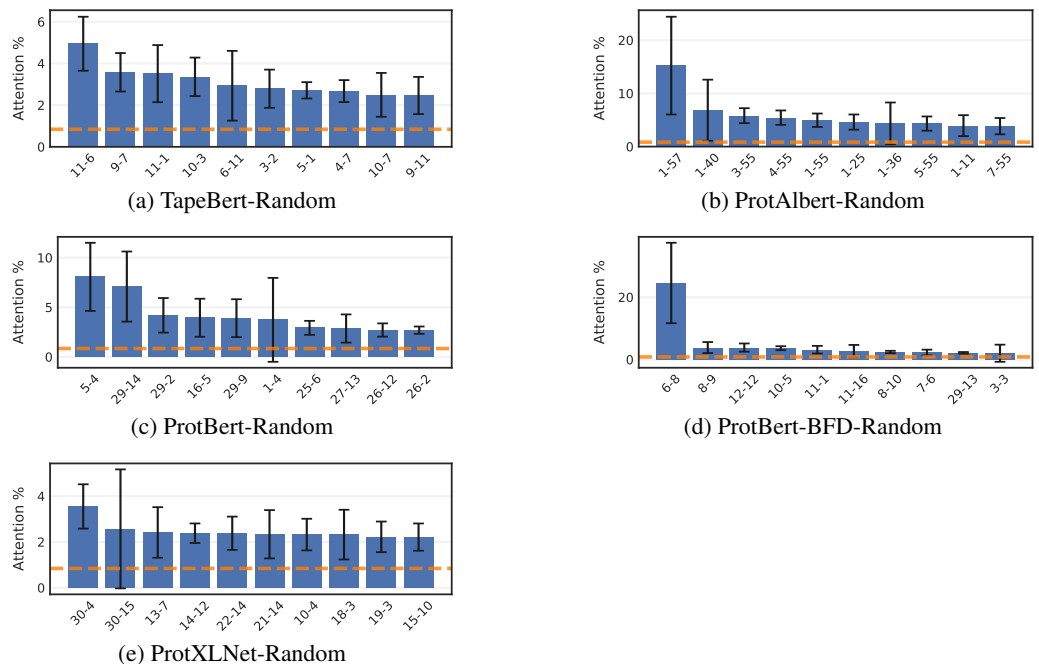

Figure 17: Top-10 heads most focused on PTMs for **null** models. See Appendix B.2 for details.

### C.5 AMINO ACIDS

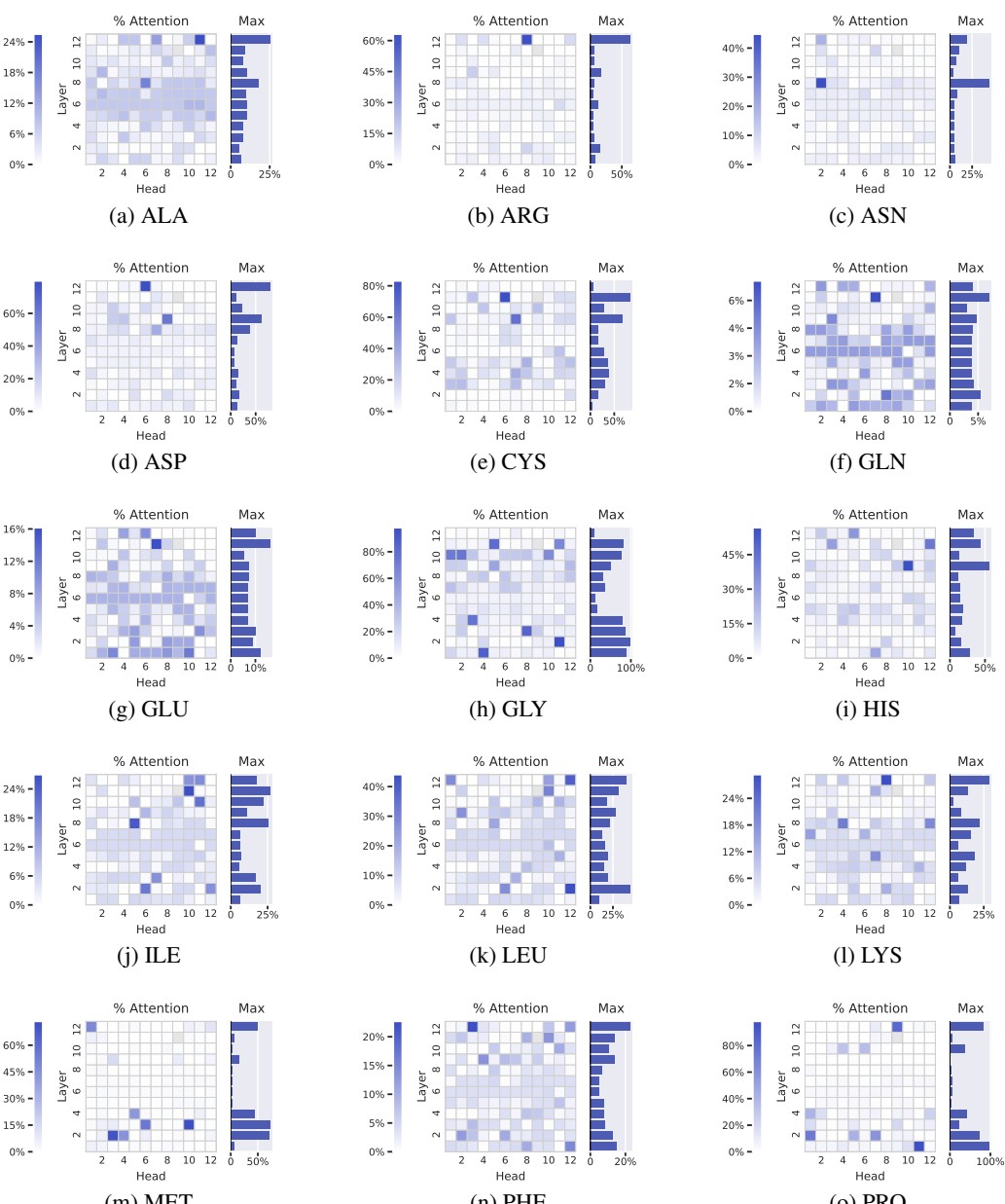

Figure 18: Percentage of each head's attention that is focused on the given amino acid, averaged over a dataset (TapeBert).

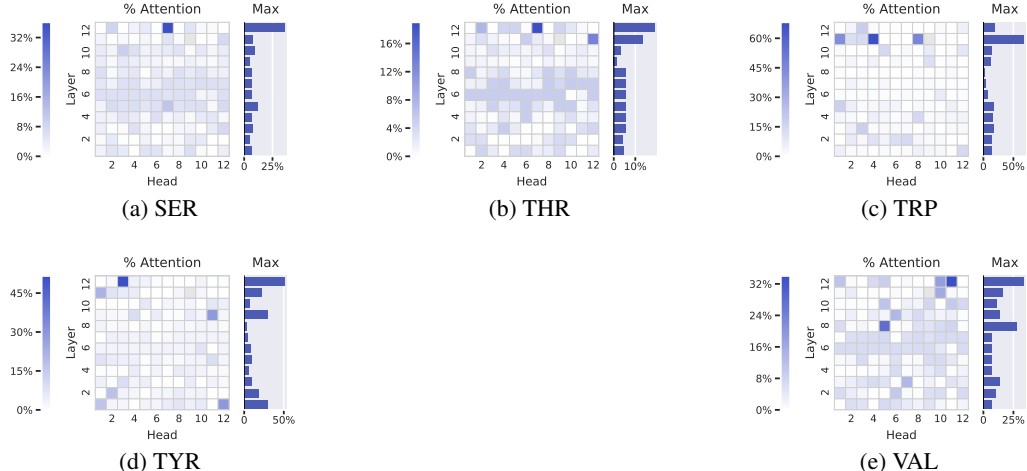

Figure 19: Percentage of each head's attention that is focused on the given amino acid, averaged over a dataset (cont.)

Table 3: Amino acids and the corresponding maximally attentive heads in the standard and randomized versions of TapeBert. The differences between the attention percentages for TapeBert and the background frequencies of each amino acid are all statistically significant ($p < 0.00001$) taking into account the Bonferroni correction. See Appendix B.2 for details. The bolded numbers represent the higher of the two values between the standard and random models. In all cases except for Glutamine, which was the amino acid with the lowest top attention proportion in the standard model (7.1), the standard TapeBert model has higher values than the randomized version.

| Abbrev | Code | Name | Background % | TapeBert | | TapeBert-Random | |
|---|---|---|---|---|---|---|---|
| | | | | Top Head | Attn % | Top Head | Attn % |
| Ala | A | Alanine | 7.9 | 12-11 | **25.5** | 11-12 | 12.1 |
| Arg | R | Arginine | 5.2 | 12-8 | **63.2** | 12-7 | 8.4 |
| Asn | N | Asparagine | 4.3 | 8-2 | **44.8** | 8-2 | 6.7 |
| Asp | D | Aspartic acid | 5.8 | 12-6 | **79.9** | 5-4 | 10.7 |
| Cys | C | Cysteine | 1.3 | 11-6 | **83.2** | 11-6 | 9.3 |
| Gln | Q | Glutamine | 3.8 | 11-7 | 7.1 | 12-1 | **9.2** |
| Glu | E | Glutamic acid | 6.9 | 11-7 | **16.2** | 11-4 | 11.8 |
| Gly | G | Glycine | 7.1 | 2-11 | **98.1** | 11-8 | 14.6 |
| His | H | Histidine | 2.7 | 9-10 | **56.7** | 11-6 | 5.4 |
| Ile | I | Isoleucine | 5.6 | 11-10 | **27.0** | 9-5 | 10.6 |
| Leu | L | Leucine | 9.4 | 2-12 | **44.1** | 12-11 | 13.9 |
| Lys | K | Lysine | 6.0 | 12-8 | **29.4** | 6-11 | 12.9 |
| Met | M | Methionine | 2.3 | 3-10 | **73.5** | 9-3 | 6.2 |
| Phe | F | Phenylalanine | 3.9 | 12-3 | **22.7** | 12-1 | 6.7 |
| Pro | P | Proline | 4.6 | 1-11 | **98.3** | 10-6 | 7.6 |
| Ser | S | Serine | 6.4 | 12-7 | **36.1** | 11-12 | 11.0 |
| Thr | T | Threonine | 5.4 | 12-7 | **19.0** | 10-4 | 9.0 |
| Trp | W | Tryptophan | 1.3 | 11-4 | **68.1** | 9-2 | 3.0 |
| Tyr | Y | Tyrosine | 3.4 | 12-3 | **51.6** | 12-11 | 6.6 |
| Val | V | Valine | 6.8 | 12-11 | **34.0** | 8-2 | 15.0 |

