# OpenReview forum: "BERTology Meets Biology: Interpreting Attention in Protein Language Models"
_ICLR.cc/2021/Conference — ICLR 2021 Poster_

### Official Review · AnonReviewer1 · 2020-10-23
**Evaluation needs to be strengthened**

**Rating:** 7
**Confidence:** 4

**Review:**

# Summary
The authors analyzed how attention and embeddings of Transformers trained on protein sequence correlate with protein properties such as pairwise contacts, binding sites, and post-translational modifications. The paper extends existing papers such as Rives 2020 ‘Biological structure and function emerge…’ by showing that layers learn more and more complex protein features with increasing layer depth and by proposing new visualization techniques. The paper is mostly clearly written while the methodological contributions are incremental. The evaluation needs to be strengthened.

# Major comments
1. The paper is missing two important baselines: 1) a Transformer with randomly initialized weights and 2) a Transformer trained on randomly shuffled protein sequences (with the same amino acid frequencies. These baselines will show if Transformers actually learn protein features, or if the correlation between network features and protein features is an artifact of the Transformer architecture. It has been shown that randomly initialized convolution networks already encode edge detectors in their weights without any training. Something similar can hold true for proteins.

2. To further support that Transformer learns protein features, I would like to see how features specialize during training on protein features. For example, how do figure 2 and figure 6 look at different numbers of training epochs?

3. The attention analysis (equation 1) depends on a threshold \theta. How was it chosen and how sensitive are results to this threshold?

4. Instead of using equation 1 for analyzing ‘alignment’ of attention $\alpha_{i, j}$ with the property $f(i, j)$, I suggest computing the Spearman correlation between  $\alpha_{i, j}$ and  $f(i, j)$ over all samples in the training set. Using the Spearman correlation is a more clear measure for quantifying the relationship between a feature and a property and does not depend on threshold.

5. Using Expected Calibration Error (ECE) for analyzing the calibration of attention is a straight-forward extension of using ECE for analyzing calibration of a ML model, which is widely performed. I also doubt the usefulness of analyzing calibration of attention since practitioners are unlikely to use a particular attention map directly for contact prediction, but train a ML model on top of attention maps to predict contacts (see Rives 2020). What matters is if the resulting ML model is calibrated.

6. For the binding site and post-translational modification (PTM) analysis (section 4.2) it is unclear how $f(i, j)$ (equation 1) is defined. A binding site and (PTM) is a property of a single position $i$ not a pair of positions $(i, j)$.

7. Section 3, datasets: How similar are test sequences to sequences that were used for training Transformer models? Sequences must not overlap and must have a maximum similarity of, e.g., 80%.

8. It is unclear which Transformer architecture was used for Figure 5 and Figure 6. For all analysis, please mention the Transformer architecture and refer to additional results for other architectures. Also discuss in each paragraph if the observation holds true for all architectures or not.

9. Section 4.2, Figure 4: I agree that TapeBert attends most strongly to the last layers but I cannot see a clear pattern for the other architectures (App C3).

# Minor comments
10. Section 3, Probing task’: Please also cite Biswas 2020 (‘Low-N protein engineering …’).

11. Section 4.1; Figure 2: ‘alignment’ can be misunderstood as ‘sequence alignment’ in the context of protein. Please use another term, e.g. ‘association’ or ‘correlation’.

12. Section 4.1: What are ‘implicit features’?

---

> ### Author Response · Authors · 2020-11-17
> **Thank you for your detailed and helpful feedback**
>
> * Thank you for your specific suggestions for baseline models. We have implemented a variation of the randomized model, as discussed in the general response. We have also explored a null model that is trained from scratch on shuffled sequences, as you proposed. In an initial training run, this model did not produce any attention weights above the threshold theta; we will report on additional results on a longer run later this week or early next week.
>
> * We agree that analyzing attention in intermediate checkpoints would be a worthwhile research direction. We are using pre-trained protein language models from public sources and unfortunately the intermediate checkpoints are not available to our knowledge.
>
> * We chose the theta value as a compromise between two concerns: choosing a sufficiently high value to make the resulting binary feature meaningful, but not choosing such a high value that we didn’t have a large enough sample size for the analysis. Besides our statistical significance testing, we limit our analysis to attention heads that have at least 100 examples (i.e. at least 100 attention weights above the threshold). Therefore we gauged the appropriateness of theta based on the number of attention heads that we could include in the analysis (i.e. that had greater than 100 associated weights). We found that a value of 0.3 resulted in the vast majority of attention heads having at least 100 examples. For example, in the analysis of contact maps, 143/144 heads in TapeBert satisfied this condition. The numbers for the other models were 460/480 (ProtBert), 465/480 (ProtBert-BFD), 747/768 (ProtAlbert), and 465/480 (ProtXLNet). Higher values of theta diminished these numbers.
>
> * With respect to sensitivity analysis of theta (0.3), we have repeated the contact map analysis (Figure 2) using thresholds of 0.2 and 0.4. The ranking of attention heads was similar to what is reported in the paper for theta=0.3, though the magnitudes of the scores differed (higher for 0.4, lower for 0.2). For the TapeBert, ProtBert, and XLNet models, the head most aligned with contacts was the same for all three theta values. For the ProtBert-BFD and ProtAlbert models, the top head for theta=0.3, was either the first or second ranking head with theta=0.2 or 0.4. Note that for these two models, there were multiple top-ranked heads with similar scores (Appendix C.2, Figure 11), so it is not surprising that the relative order might have changed.
>
> * Spearman rank correlation is an interesting idea, however we were not certain how to apply it in the case of a binary variable representing the protein properties (binding sites, contact maps, etc.).
>
> * We have moved the attention calibration discussion to Appendix D and discussed limitations; please see additional details in the general response.
>
> * We have changed the language following Equation 1 to clarify how it may be applied to properties of a single amino acid (e.g. PTMs and binding sites). Essentially we define f(i,j) to be 1 if the property is present at position j.
>
> * We use the train/validation splits from two existing datasets from the tape repository: the ProteinNet dataset and the Secondary Structure dataset.  The splits are filtered at 30% and 25% sequence identity, respectively (see Rao et al. 2019 for more details).
>
> * Figures 5 and 6 refer to the TapeBERT model. We make this explicit in the new version and ensure that any additional results for other models are explicitly mentioned.
>
> * We have amended the statement that binding sites are targeted in the deepest layers.
>
> * Thank you for the additional citation; we have updated the related work.
>
> * We no longer use the term “alignment” to refer to the correspondence between attention and properties (though we use the forms “align” and “aligned” in some places).
>
> * We have changed “implicit features” to “properties” (Appendix D).

---

> > ### Author Response · Authors · 2020-11-20
> > **Update on null model trained on shuffled sequences**
> >
> > We have completed a full training epoch for the null model trained on the randomly shuffled sequences. This model appears to have converged to focusing attention roughly evenly across all positions in the sequence. As a result, none of the attention weights are above the minimum threshold theta set in our analysis for any of the protein sequences in the dataset. This suggests that the high-confidence (>theta) attention weights themselves are not due to chance, echoing the results from the randomly initialized model.
> >
> > We have updated Section B.2 to reflect this result in the latest uploaded draft. We believe that the null model introduced in the initial revision, which shuffles attention weights from a trained model, will provide a reasonable baseline for evaluation, in combination with the tests of statistical significance.

---

### Official Review · AnonReviewer2 · 2020-10-28
**Good paper, interesting topic/application of attention but explanatory goals is only weakly supported**

**Rating:** 6
**Confidence:** 4

**Review:**

General comments:

1) authors claim

"We note that all of the above analyses are purely associative and do not attempt to establish a causal link between attention and model behavior (Vig et al., 2020; Grimsley et al., 2020), nor to explain model predictions”.

and in the related work section

"In our work, we take an interpretability-first perspective to focus on the internal model representations, specifically attention and intermediate hidden states, across multiple protein language models. We also explore novel biological properties including binding sites and post-translational modifications.”

However the structure of evidence is very phenomenological in nature — x (attention at head H, layer L) is observed which coincides with phenomena Y (binding/PTM/).


2) no cross-model analysis. Why are we observing different outcomes in different models?

Specific comments:

section 4.1 "Considering the model was trained on a masked language modeling task without any spatial information, the presence of these structurally-aware attention heads is intriguing”
	- why strongest alignment in only few heads and in high layers?
	- "While there seems to be a strong correlation between the attention head output and classically-defined contacts, there are also differences”. What are the differences?

	- Attention as a well-Calibrated predictor section. Authors offer no explanation why the ECE is higher for none TapeBert.

section 4.2 *why* is the causal reason for attention focusing on binding sites or PTMs?

section 4.4: what about the other 4 amino acids that dont have 25% allocated attention in TapeBert? what is it about those amino acids?

---

> ### Author Response · Authors · 2020-11-17
> **Thank you for your thoughtful feedback**
>
> * We agree that the results are phenomenological, and we have attempted to convey this in stating that the analysis is associative rather than causal. We are open to suggestions on how we may improve this language.
>
> * As mentioned in the general response, we have moved additional cross-model results for binding sites from the appendix to the body of the paper (Figure 3) and added additional discussion points around the observed differences.
>
> * With respect to differences between attention head outputs and classically defined contacts, we observed cases where (1) the attention targeted amino acids that did not satisfy the formal definition of contacts (Sec. 2) and (2) attention did not target amino acids that were defined as contacts. We believe the first case may be due to the rigid definition of contacts, which does not consider amino acids pairs that lie outside of the specified distance thresholds but still chemically interact with one another. The second case may identify amino acid pairs that are formally defined as contacts, but do not interact strongly with one another.
>
> * As mentioned in the general response, we have moved the attention calibration section to Appendix D. It is unclear why the TapeBert model performs better than the others. It may be coincidental, though more investigation is needed.
>
> * With respect to the causal mechanism for targeting binding sites (and by extension, PTMs), we have added a final sentence to the fourth paragraph of section 4.2 to clarify this. The section on PTMs points back to this updated section.
>
> * The 4 amino acids that did not receive at least 25% of attention from at least one head are: Gln (7%), Glu (16%) Phe (22%), and Thr (19%) (see Appendix C.5, Table 2). We note that these percentages are nonetheless higher than the background percentages of these amino acids by a statistically significant margin. In all cases except for Gln, the results of the null model were lower than that of the standard model (see same table).  Thus 3 of the 4 amino acids support the general result; more investigation is needed to determine why Gln exhibits different behavior in this case.

---

> > ### Author Response · Authors · 2020-11-24
> > **Additional cross-model discussion**
> >
> > As mentioned in the general response, we have added a brief discussion to 4.1 on differences between the 5 pre-trained models concerning attention to contacts (in addition to previously added discussion around binding sites), as well as a table in Appendix A.1 that summarizes the properties of the models.

---

### Official Review · AnonReviewer3 · 2020-10-29
**Cool paper but: a little loose with biological concepts and a major statistical concern**

**Rating:** 7
**Confidence:** 4

**Review:**

This is paper is an investigation of the potential biological interpretation of attention in five pretrained protein sequence Transformer models (one from TAPE and four from ProtTrans). It seems that the meatiest contribution in this paper involves the use of the attention analysis metric p_alpha(f) to measure the co-occurrence of attention values (above threshold alpha) and a contact matrix f. A related contribution is the definition of attention calibration, which is a slight variant of Expected Calibration Error applied to evaluating the calibration between attention values and a discretized contact matrix.

My biggest concern with this paper is that they analyze Transformer models by looking for a single attention head that maximizes p_alpha(f) without, as far as I can tell, really evaluating how often these multiple comparisons might lead to a high p_alpha(f) value by chance. Several compelling examples are shown for individual attention heads (e.g. repeated evaluation of TapeBert 12-4). These individual heads really could just be the locus in which a pretrained model encodes understanding of structural adjacency but...there has to be a null model when trying every node of a big neural network and taking the max value of an evaluation metric. The paper repeatedly plots the background frequency of contacts between residues in the protein data but that's not the same as determining how often your analysis would yield high values just by chance.

Similarly, what's the null model for the calibration analysis? The plot, like others, has an orange line showing the background frequency of contacts but that seems really unrelated to calibration. I don't know what the null model *should* be but this paper should endeavor to show that finding one calibrated attention head (among the many possibilities) is surprising.

Other comments:

* In the first background items, on "Amino acids", the description derails into discussion of substitution matrices. If understanding substitution matrices is important to the paper, give them their own bullet point. Otherwise it's a strangely specific technique to discuss while describing amino acids in general.

* "every protein sequence is formed from a vocabulary of 20 standard amino acids" it's strange to say this and then talk about PTMs a few paragraphs down. If you're presenting PTMs to the networks you're evaluating then you're likely using a larger amino acid alphabet (I think TapeBert has 25 amino acids?)

* This description of secondary structure seems odd: "Secondary structure describes the local segments of proteins, and may be divided into three broad categories: Helix, Strand, and Turn/Fold, as well as an Other category for local structure that falls outside these categories." If you asked most biologists what the broad categories of secondary structure are would probably say "[alpha] Helix" "[beta] sheet", it didn't even occur to me to break sheets into strands and turns. I guess it makes sense that these categories might occur in some secondary structure prediction datasets or tasks, but it seems like the details of some dataset are being presented as representative of biology.

* Slightly confusing terminology here: "Binding sites are protein regions that bind with other molecules to carry out a specific function. For example, the HIV-1 protease is an enzyme responsible for a critical process in replication of HIV (Brik & Wong, 2003). It has a binding site, shown in Figure 1b, that is a target for drug development to ensure inhibition." I think it's important to distinguish the active sites of a protein where that protein binds some natural target or ligand and the binding site of a drug which inhibits the protein. These often overlap to some degree but the binding site of a small molecule inhibitor can be a much smaller pocket than a natural ligand.

* "Attention targets PTMs in a small number of heads": I wish it was clearer what this analysis meant. Were the input sequences given as unmodified AAs and then you're evaluating which residues are often observed as modified in vivo? Or did you give the networks sequences with PTMs in them and then this attention pays attention to when rare AAs get used?

OK, after this list of complaints, I want to come back to what I like about this paper. I think many people (myself included) suspect that attention in protein Transformer language models operates in biologically interpretable ways. It's really nice to actually show it and also show that attention is compatible with other simpler biological notions (like AA substitution matrices). But: it's not enough to just look for the best node in each network, you have to make some kind of effort to quantify how like that finding would be by chance in a large enough network.

---

> ### Author Response · Authors · 2020-11-17
> **Thank you for your detailed feedback and suggestions**
>
> * With respect to your statistical concern, we have made two changes. First, we have included a null model as described in the general response. Second, we have incorporated the Bonferroni correction into our estimates of statistical significance, as also discussed in the general response. Both are described in detail in Appendix B.2.
>
> * We have moved the calibration analysis to Appendix D and included discussions of the limitations/assumptions, as mentioned in the general response.
>
> * With respect to the background section [2], we have assigned a dedicated bullet to the substitution matrix as suggested. We have also changed the categories of secondary structure to beta sheet and alpha helix. In the Methods section (3, Datasets) we distinguish these categories from the more fine-grained categories used in the analysis, and provide some motivation for this taxonomy.
>
> * To answer your question, the model is not trained on information regarding PTMs; the datasets are limited to the standard IUPAC codes. In our evaluation, we examine how attention can recover patterns to the residue positions in which PTMs occur.  We’ve changed the language slightly in section 4.2 to clarify that we are analyzing the positions where PTMs would occur.
>
> * With respect to binding sites, the position of the binding site is curated through RCSB PDB and involves binding to other proteins, natural ligands, or small-molecule drugs. We have updated Section 2 to reflect this. In the case of 7HVP, the active site is a limited region known to bind to both natural proteins and designed inhibitors.

---

### Official Review · AnonReviewer4 · 2020-10-29
**Systematic analysis of of interpretability of Transformer-based language attention models used for protein sequences**

**Rating:** 6
**Confidence:** 4

**Review:**

The authors of this manuscript conducted a comprehensive analysis of the interpretability of self-attention language models when learning from protein sequences. Specifically, five multi-head self-attention models from NLP were used to model protein sequences. The following tasks were conducted: (1) whether attention captures secondary and tertiary structural information, (2) how and why attention targets binding sites and post-translational modifications, (3) how attention captures properties of varying complexity across different encoding layers, and (4) relationship between pairwise attention amino acids and substitution matrix. Through measures and visualizations, the authors can discover interesting relations between attention and various structural properties. In general, I think it is a good analysis paper. Even though there is no contributions in modelling, the authors can use suitable tools to probe and show the relationships between attention and protein structural informations. I believe the findings of this work is informative and inspiring to researchers in bioinformatics.

This paper can be enhanced if consistency and difference between these five attentions models can be further analyzed in the four tasks. What would be the most suitable model for protein sequences?

Moreover, I would suggest the authors to discuss how many layers and heads are needed for protein sequence modelling. What will happen if 50 or just 5 layers are used for the analysis in this paper? Will the same conclusions be drawn?

Minors:
1. transformer -> Transformer
2. The list of references can be further formatted. For example, "Advances in neural information processing systems" -> "Advances in Neural Information Processing Systems".

---

> ### Author Response · Authors · 2020-11-17
> **Thank you for your thoughtful feedback**
>
> * As discussed in the general response, we have moved the cross-model results for binding sites from the appendix to the main body (Figure 3) and included additional discussion around the differences between the models.
>
> * We agree that further analysis of the number of layers and heads would be helpful, though we would leave this to future work given time constraints. We note that the models studied vary in the number of layers (12 to 30) and heads (12 to 64), so this provides some indication that the observed behavior may be invariant to model size.
>
> * Thank you for pointing out the formatting issues. We have resolved these in the updated manuscript.

---

> > ### Author Response · Authors · 2020-11-23
> > **Additional cross-model discussion**
> >
> > As mentioned in the general response, we have added a brief discussion to 4.1 on differences between the 5 pre-trained models concerning attention to contacts, as well as a table in Appendix A.1 that summarizes the properties of the models.

---

### Official Review · AnonReviewer5 · 2020-11-06
**Good paper but for "calibration"**

**Rating:** 7
**Confidence:** 4

**Review:**

### Summary

This paper offers an in-depth analysis of attention in large-scale language models including (AL)BERT and XLNet. Key "headline" findings:

- Attention maps highlight amino acid pairs that are far in sequence space but near in coordinate space.
- Attention maps highlight binding sites within proteins.
- Attention maps capture local secondary structure.
- Attention heads have affinities to specific residue types.

More nuanced findings:

- The evolution across layers of attention-encoded information differs from embedding-encoded information: latter accumulates steadily while former can be isolated to single (end) layer.
- Evidence indicates that attention heads' affinities to specific residue types rests on physical properties rather than mere identity.

### Merits

This is an interesting study that is modest in its characterization of its contributions but diligent in its empirics. Too many papers in this space try to train giant models and cherry-pick tasks for which those models "beat" non-neural baselines, while contributing too little (in the reviewer's humble opinion) to our understanding of what's going on. The authors have shed light on the attention mechanism, showing the intuitive but notable results that attention correlates with coordinate-space distance, binding sites, secondary structure, and residue identity.

I particularly liked the authors' experimental design in 4.4 to shed light on whether identity or structural/functional properties are driving the attention.

The paper is generally well-written and will be accessible to those without prior exposure to protein modeling. The authors' efforts toward accessibility are clear both in the length of background exposition and in the terminology: although I use the term "residue" in this review, I appreciate the authors' choice of "amino acid".

### Drawbacks

**Calibration**: I had to read the "Attention is a well-calibrated predictor of contact maps in some cases" section about 10 times over before I had a grasp on what was going on. Even with my current understanding of the section, I'm extremely skeptical about this specific study. Why do I expect attention (which produces a point on the simplex) to be at all calibrated to a contact map (which is an arbitrary pairwise zero-one matrix)? Suppose we had a very tightly clustered protein and a high-enough radius such that all residues were considered to be in contact. No possible attention map would be well-calibrated to the ground-truth contacts.

I would recommend removal of this section. It is a weak bordering on nonsensical result which seems to have been plucked from a single attention head. I suspect this result is not robust and in any case it does not contribute much conceptual understanding.

**Embeddings vs. attention**: In related work, the paper cites prior work that analyze the output embeddings of large scale models for structural and functional properties. The paper itself focuses on properties captured through attention. As a reader, I'm left somewhat unsatiated. What's captured in embeddings that attention represents less well? What's captured in attention that might not necessarily emerge through a metric space defined in embedding land? The paper missed an opportunity in my opinion to do a systematic comparison of information encoding in attention vs. embeddings, aside from what's shown in Figure 6.

### Recommendation

This paper does not make, nor claim to make, novel methodological contributions in the modeling of proteins. What it does is to study Transformer attention using previously-known techniques but applied to protein sequence models. It overall performs an admirable job of doing so. We at ICLR care not only about "learning representations", but about assessing what precisely those representations have learned, so the paper is certainly appropriate for ICLR.

My main hesitation with recommending acceptance is the whole attention calibration business. The metric is mathematically unsound (contact maps certainly do not sum either row-wise or column-wise to one) and the fact that this paper found a "well-calibrated" attention head isn't all that notable.

With that caveat, I cautiously recommend acceptance. I will raise my score upon either removal of "attention calibration" from the paper or an *extremely* convincing explanation of why I've misinterpreted the metric / why I should care about the calibration of an attention map with an arbitrary unnormalized zero-one matrix.

UPDATE: The authors have removed the attention calibration study. My new score is a 7. This is a good paper and I heartily recommend its acceptance.

### Comments

- p1: "substitution properties" is a handwavy term. Please make the meaning more precise.
- p3: "We note that all of the above analyses are purely associative and do not attempt to establish a causal link between attention and model behavior (Vig et al., 2020; Grimsley et al., 2020), nor to explain model predictions (Jain & Wallace, 2019; Wiegreffe & Pinter, 2019)." very good thank you!
- p4: very nice ALBERT visualization and consistent with the model architecture!
- p6: "deeper layers focus relatively more attention on binding sites and contacts". OK, contacts sure, but binding sites?

---

> ### Author Response · Authors · 2020-11-17
> **Thank you for your detailed feedback and suggestions for improvement.**
>
> * As mentioned in the general response, we have moved the attention calibration section to Appendix D, and we have removed related claims from the body of the paper. We acknowledge your point that attention weights are normalized to sum to one while ground-truth contact maps are not. The unstated premise of this analysis is that contacts are sparse, that each amino acid in practice only has one or a small number of contacts. We add a figure that shows the degree to which this assumption holds (see Appendix D, Figure 21), and include discussion around this limitation. Please let us know if this sufficiently addresses your concerns.
>
> * We agree that further comparative analysis of embeddings and attention weights is a promising research direction. We leave this to future work given time constraints.
>
> * We have replaced “substitution properties” with a more explicit phrasing.
>
> * We have corrected the characterization that binding sites are targeted in the deepest layers.

---

> > ### Comment · AnonReviewer5 · 2020-11-19
> > **Please go further**
> >
> > I thank the authors for their revision. My original score stands.
> >
> > As the data reveals in Figure 21, the "calibration" view is only appropriate about 10% of the time and completely arbitrary the other 90% of the time. I appreciate what the authors are trying to show (roughly, "attn weight \propto likelihood of contact") but the present result is unsuccessful in doing so.
> >
> > I *strongly* encourage the authors to just remove Appendix D entirely; I am sure none of my fellow reviewers will miss it terribly. My score would be a 7 upon removal.

---

> > > ### Author Response · Authors · 2020-11-20
> > > **Appendix D removed**
> > >
> > > Thank you for the followup response. We understand your point and have removed Appendix D in the latest uploaded draft.

---

> > > > ### Comment · AnonReviewer5 · 2020-11-20
> > > > **Update**
> > > >
> > > > I have updated my review accordingly. Thanks!

---

### Author Response · Authors · 2020-11-17
**Collective response to reviewers: revised draft uploaded**

We are grateful to the reviewers for taking the time to share their detailed feedback and suggestions, which we found very helpful for improving the paper. We have uploaded a revised draft based on these suggestions. We summarize the major changes below, and respond to specific points in the individual threads.

* Null model: Multiple reviewers suggested incorporating a null model in the analysis. While we do include statistical significance tests relative to the background frequency of the properties analyzed [C.2, C.3, C.4, C.5 ], we agree that it is also important to include a null model as a baseline.  As suggested by reviewers, we evaluated randomly initialized versions of each of the models; however, these did not yield any attention weights above the specified threshold theta. Therefore, we implemented an alternative randomization scheme in which we randomly shuffle attention weights from the original models as a post-processing step. As now described in detail in Appendix B.2, we shuffle the weights such that the resulting attention distributions are still valid. We now include the results from these models in Appendices C.2, C.3, C.4, and C.5, and at the end of Section 4.4.

* Statistical significance testing: in response to the concern that some results may be spurious due to aggregating the metrics over many attention heads, we have incorporated the Bonferroni correction [B.2], which accounts for this case of multiple hypotheses corresponding to multiple heads [C.2, C.3, C.4, C.5].

* Attention calibration: We have moved this section to Appendix D, including additional discussion around the points raised. We address specific points in our responses to reviewers.

* We have moved the full cross-model results for binding sites to the body of the paper (Figure 3) and added discussion points around differences between the models.

* We have standardized the discussion and language around biological concepts.

---

> ### Author Response · Authors · 2020-11-20
> **Additional revision**
>
> Based on additional feedback, we have removed Appendix D (attention calibration) from the latest draft. We believe this change is consistent with the original feedback received from multiple reviewers. We have also added a brief discussion around a second null model to Appendix B.2 based on AnonReviewer1's proposal.

---

> ### Author Response · Authors · 2020-11-23
> **Updated draft**
>
> We’ve added a brief discussion to Section 4.1 on differences in results between the 5 models with respect to contact maps,  and also added a new summary table to Appendix A.1 comparing the properties of the 5 models. We also made some minor wording changes and shortened some figure captions.

---

### Decision · Program_Chairs · 2021-01-07
**Final Decision**

**Decision:**

Accept (Poster)

**Comment:**

This paper offers an in-depth analysis of attention in large-scale language models including (AL)BERT and XLNet in the context protein representation learning, and obtains many interesting findings. This is not a typical paper with novel technologies proposed, instead, it studies the existing technologies in a specific (biology) context and explains what the learned representations and attention map really mean.
All the reviewers see the value in this paper and give positive feedback in general. At the same time, they also raised a few concerns, e.g., regarding the claim on “well-calibrated" attention head, on some missing details of the algorithm description and the experiments,  on phenomenon vs. causality of the finding, etc. The authors really did a very good job in their rebuttal and paper revision, and most of these concerns were (at least partially) addressed, and a few reviewers raised their scores. With this, we are quite confident that this paper is above the bar of ICLR.